# Fault Tolerant Control Based on an Observer on PI Servo Design for a High-Speed Automation Machine

**Prathan Chommuangpuck [1,2], Thanasak Wanglomklang [1], Suradet Tantrairatn [1] and Jiraphon Srisertpol [1,\*]**

[1] School of Mechanical Engineering, Suranaree University of Technology, Nakhon Ratchasima 30000, Thailand; prathan8137@hotmail.com (P.C.); thanasak.wang@gmail.com (T.W.); suradetj@sut.ac.th (S.T.)

[2] Western Digital (Thailand) Co., Ltd., Bang Pa-in Industrial Estate, Phra Nakhon Si Ayutthaya 13160, Thailand

[\*] Correspondence: jiraphon@sut.ac.th; Tel.: +66-4422-4412

**Abstract:** The fault tolerant control (FTC) technique is widely used in many industries to provide tolerance to systems so that they can operate when a system fault occurs. This paper presents a technique for FTC based on the observer signal application, which is used for a high-speed auto core adhesion mounting machine. The utilization of the observer signal information of the linear encoder fault is employed to adjust the gain parameters to achieve the appropriate gain value while maintaining the required performance of the system. The dynamic modeling of the servo motor system design utilizing a pole placement technique was designed to support the proposed method. A scaling gain fault step size adjustment from −1% to 1% with increments of 0.2% is used to simulate the fault conditions of the linear encoder. The statistical mean value of the observer error signal is used to train the artificial neural network (ANN) model. The results showed that the control system design successfully tracked the dynamic response. Furthermore, the ANN model, with more than 98% confidence, was satisfactory in classifying the linear encoder fault condition. The gain compensation was successful in reducing position error by more than 95% compared with the system without compensated gain.

**Keywords:** servo system design; fault tolerant control; artificial neural network; fault detection and isolation; observer design

## 1. Introduction

A hard disk drive (HDD) is a storage data device using a magnetic recording head assembled into a head gimbal assembly (HGA) which writes and reads data to/from the disk. The HGA shown in Figure 1 consists of two major components, which are the suspension and slider. The auto core adhesion mounting machine (ACAM) shown in Figure 2 is used for adhesive dispensing and slider attachment onto the suspension, requiring the positioning of the adhesive and slider attachment at the sub-micron level. The machine uses a feed drive actuator to move the worktable, as shown in Figure 3, moving the suspension to the desired position for dispensing and attachment. The linear bearing is used to support the worktable in the feed drive actuator and the linear encoder is used for position checking and feedback to the motor controller. The machine operates continuously for 24 h and meets the high-speed and high-reliability requirements. A degeneration or fault of the linear encoder causes reduces the performance to control the worktable position and results in unplanned downtime. The mispositioning of the reference holes for adhesive dispensing and slider attachment, as shown in Figure 4, results in an incorrect position for dispensing the adhesive on the suspension, as shown in Figure 5. Thus, a preliminary fault of the linear encoder must be detected before the

machine generates a defect, and the fault tolerant control of the linear encoder is necessary to ensure that the machine runs with the desired performance.

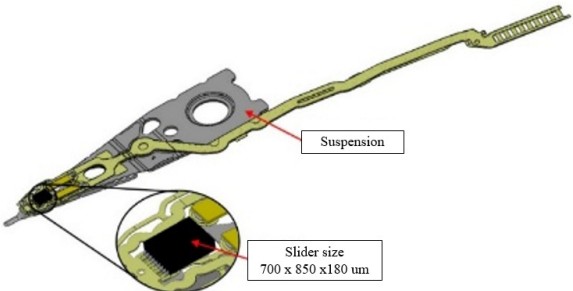

**Figure 1.** Head gimbal assembly (HGA) component.

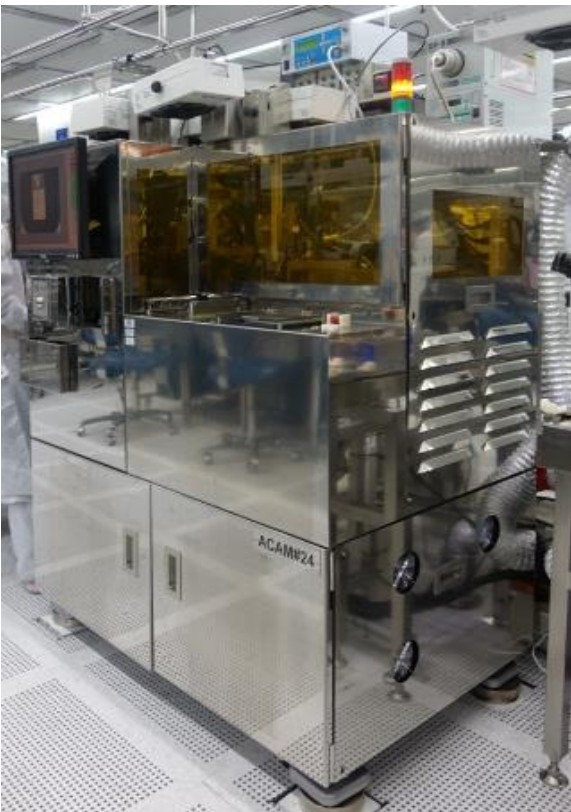

**Figure 2.** Auto core adhesion mounting machine.

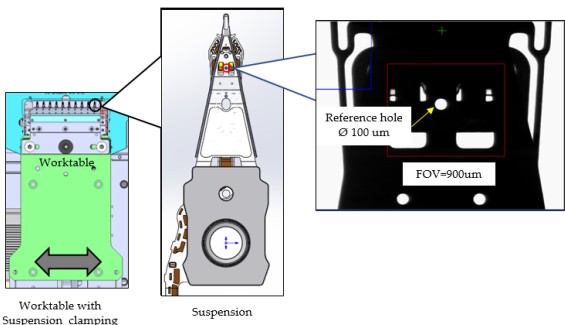

**Figure 3.** The worktable with suspension clamping

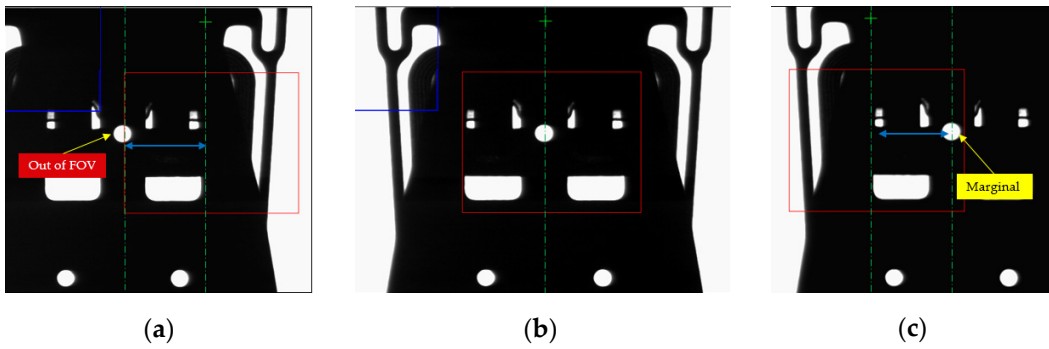

|     |     |     |
| :-: | :-: | :-: |
| (**a**) | (**b**) | (**c**) |

**Figure 4.** Reference hole positions: (**a**) fault condition, (**b**) nominal position, (**c**) marginal position.

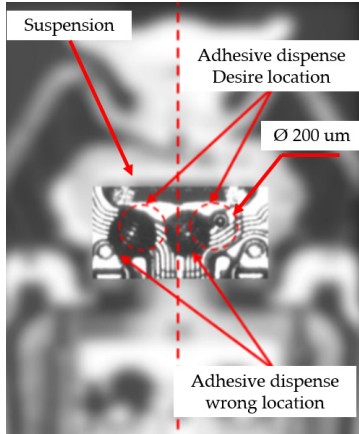

**Figure 5.** Adhesive dispensing in the wrong position.

The approaches to the design of the controller for the actuator, fault detection, and fault tolerant control were reviewed as follows. Saengsri Sri compared the use of a PI-servo with Okata's method. The experiment evaluated the durability of external noise [1]. The researcher developed the controller for a state-level observer with the pole placement method for an electromagnetic wave with the servo system [2]. Xie D. proposed a control system with fuzzy logic to control the feed drives in a CNC machine [3]. The performance of the friction model was improved by the PID and status feedback control and extended to the XY feed drives [4]. Besides this, bearing faults were considered an important component. The classification of ANN using vibration signals from Hilbert footprints was used for analysis; therefore, the results show that the proposed method achieves 87.3–100% accuracy [5]. Furthermore, bearing error detection using a deep artificial neural network was proposed by Zhao D. A method of automatic recognition in time-domain images, which achieved 98.3% accuracy, was proposed in [6]. On the other hand, Chenxi studied the diagnosis of rolling bearing faults by providing a deep neural network for classification. The four properties that are applied to the vibration signal are important information both in time and frequency domains [7]. The fault detection and diagnostics of the linear bearing by an artificial neural network (ANN) was proposed. The accuracy reached more than 93% using the data set of the motor current, FFT, and crest factor. In [8], a method was developed for the FTC to compensate for both malfunctions of the actuator and the sensor. Actuator faults including a loss of system performance interfered with this. In minor control laws, the existence of compilers on controllers may only compensate for constant errors [9]. The article presented a functional error control system (AFTCS) for fuel cells/hybrid power transmission systems for batteries used in city buses. AFTCS consists of systems for detecting and diagnosing faults and controllers; it could be configured again as an algorithm for detecting and isolating three important defects that had been presented. Real-time adjustable controllers were utilized to maximize the efficiency of the pre-fault system. The experimental results achieved the proposed system performance [10]. The fault

detection of a linear bearing by ANN based on observer information was also proposed. The dynamic model was obtained by the pole placement technique. An experiment to simulate the detection of linear bearing conditions was conducted. The result showed that the ANN model reached 99.7% accuracy [11]. The article introduced the PSO algorithm to tune the CNC servo system. The PSO algorithm was used to obtain the optimal controller parameters. The simulation and experiment of two-step servo tuning showed the effectiveness of this approach [12]. Indoor air quality (IAQ) monitoring by ANN pattern recognition was proposed. The nine sensors collected air quality data and transmitted these to the base station through a wireless system. The ANN pattern recognition was used to classify air quality. The results showed that the proposed system was capable of measuring and successfully classifying the IAQ in various environments [13]. An artificial neural network (ANN) was built to generate a user-friendly mathematical expression to examine the relationship between the NPV of PVs and specific factors of interest and to investigate the potential of utilizing photovoltaics for electricity generation under the meteorological and working conditions of Landskrona. The result showed that the neural network was able to efficiently utilize model PV systems with a coefficient of determination close to that provided in [14]. The prevention of failures is vital in complex rotating machines in industrial operations. The methodology applied to the two-stage artificial neural network (ANN) classification approach could pave the way for the automatic classification of rotating machine faults, allowing us to accurately detect and classify rotor-related anomalies to achieve truly robust maintenance decision-making systems. The results at both stages of the algorithm showed promising potential for enhancements during the condition monitoring of critical machines, as presented in [15]. The detection and classification of wood based on the artificial neural network (ANN) algorithm used backpropagation and the conjugate gradient method in the training process. The ANN was able to improve the accuracy of the process by up to 96.42% with an optimal 0.2 learning rate, 200 hidden layers, and 100 epochs for the detection of types and for identification classification in [16]. A large off-shore wind turbine represents a promising source of emission-free electricity; this paper applies fault detection and diagnosis (FDD) and an active fault tolerant control system (AFTCS) to improve the regulation of the generator speed. The fuzzy gain schedule (FGS) technique is used to improve the PI controller. The active fault tolerance capabilities received the information from the FDD system based on the signal correction technique. The simulation result showed the effectiveness of the proposed method for both the fault-free and faulty conditions [17]. The ANN model fitting with the Levenberg–Marquardt (LM) algorithm was used to predict the three outputs of the MEMS Cantilever by the five input parameters that were obtained from MEMS simulation software. The ANN model fitting result showed a perfect fit of the data between the input and output, and the R square value being close to 1 indicated a reasonably good fit [18]. An observer-based gain scheduling controller for the flight of an unmanned helicopter was designed by linear quadratic integral controllers for two linear operating points. Then, observer-based gain scheduling was utilized to blend the individual controllers. The result showed satisfactory performance [19]. A state derivative feedback robust gain scheduling method was proposed for the stabilization of an LTI system. The D-stability methodology was used to improve the performance of the transitory response. The result showed that the method could successfully increase or decrease controller gain [20].

This article presents the design and development of a fault tolerant control system for a linear encoder based on the observer signal information used for a high-speed ACAM machine. The PI servo system was designed and developed based on the pole placement technique with a state observer. The observer error from the observer signal is used for sensor fault classification by ANN. The result of the classification of the observer error signal applied feedback to adjust the mechanism of the controller by selecting the appropriate gain value to compensate and sustain the system under the desired conditions. The experiment was conducted in two phases: the first phase was to validate and compare the performance of fault detection and diagnostics using ANN between pattern recognition and the model fitting method, and the second experiment was performed to validate the gain compensation.

Finally, we compared the performance of continuous gain scheduling and discrete gain scheduling with the system without gain compensation.

The objective of this research is to develop a gain scheduling application to compensate the gain value, adding tolerance to the system and allowing it to run with the desired performance when a sensor fault is detected. We use the proposed technique to prevent a position error of the suspension in the worktable of the ACAM machine. The remainder of this article is structured as follows: Section 2 outlines the dynamics modeling of the feed drive with a DC servomotor and the architecture of the fault tolerant control. Section 3 discusses the experimental setup. The results and conclusions are discussed in Sections 4 and 5.

## 2. Materials and Methods

### 2.1. Dynamics Modelling of Feed Drive with DC Servomotor

The mock-up of the ACAM machine was designed and developed for the proposed experiment; the system included an *x* and *y*-axis driven by the servo motor and coupled with the lead screw. The work table was mounted on top of the four linear bearings. The linear encoder was used to check and feedback the position of the motor to the controller. A redundant rotary encoder was installed on the other side of the lead screw to double-check the position of the clamping unit, as shown in Figure 6a [11]. This work aimed to increase the system precision of the *x*-axis. The modeling of the single-axis is presented in Figure 6b [12]. The system's equations assume that the lead screw is a rigid body and were analyzed for both electrical and mechanical integrity according to Equations (1)–(5).

$$i_a(t) = \frac{1}{L_a}V_a(t) - \frac{R_a}{L_a}i_a(t) - \frac{K_b}{L_a}\theta_m(t) \tag{1}$$

$$\ddot{\theta}_m(t) = -\frac{B_m}{J_m}\theta_m(t) + \frac{K_t}{J_m}i_a(t) \tag{2}$$

$$\ddot{x}_t(t) = -\frac{C_t}{M_t}x_t(t) - \frac{K_s}{M_t}x_t(t) + \frac{RK_s}{M_t}\theta_m(t) \tag{3}$$

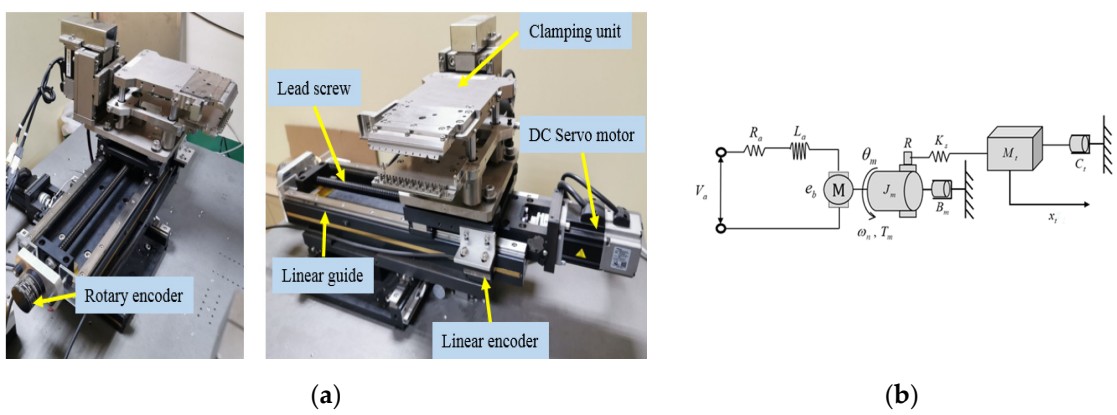

(**a**)               (**b**)

**Figure 6.** (**a**) Mockup unit of linear stage and motor; (**b**) physical modelling of the linear stage servo motor.

According to Equations (1)–(3), the state-space model and the state vector $x(t) = [\ i_a(t)\ \ \theta_m(t)\ \ \theta_m(t)\ \ x_t(t)\ \ x_t(t)\ ]^T$ are arranged in matrix form, as shown in Equations (4)–(5).

$$
\frac{d}{dt}\begin{bmatrix} i_a \\ \theta_m \\ \theta_m \\ x_t \\ x_t \end{bmatrix} = \begin{bmatrix} -\frac{R_a}{L_a} & 0 & -\frac{K_b}{L_a} & 0 & 0 \\ 0 & 0 & 1 & 0 & 0 \\ \frac{K_t}{J_m} & 0 & -\frac{B_m}{J_m} & 0 & 0 \\ 0 & 0 & 0 & 0 & 1 \\ 0 & \frac{RK_s}{M_t} & 0 & -\frac{K_s}{M_t} & -\frac{C_t}{M_t} \end{bmatrix}\cdot\begin{bmatrix} i_a \\ \theta_m \\ \theta_m \\ x_t \\ x_t \end{bmatrix} + \begin{bmatrix} \frac{1}{L_a} \\ 0 \\ 0 \\ 0 \\ 0 \end{bmatrix}u \tag{4}
$$

$$
y(t) = \begin{bmatrix} 0 & 0 & 0 & 1 & 0 \end{bmatrix}\begin{bmatrix} i_a & \theta_m & \theta_m & x_t & x_t \end{bmatrix}^T \tag{5}
$$

$$
A = \begin{bmatrix} -\frac{R_a}{L_a} & 0 & -\frac{K_b}{L_a} & 0 & 0 \\ 0 & 0 & 1 & 0 & 0 \\ \frac{K_t}{J_m} & 0 & -\frac{B_m}{J_m} & 0 & 0 \\ 0 & 0 & 0 & 0 & 1 \\ 0 & \frac{RK_s}{M_t} & 0 & -\frac{K_s}{M_t} & -\frac{C_t}{M_t} \end{bmatrix}, B = \begin{bmatrix} \frac{1}{L_a} \\ 0 \\ 0 \\ 0 \\ 0 \end{bmatrix}, C = \begin{bmatrix} 0 & 0 & 0 & 1 & 0 \end{bmatrix}
$$

In this research, the system identification method takes advantage of the estimation of the system parameters of the feed-driven system, which is considered experimental data. The results are described in Table 1.

**Table 1.** Parameters of the feed-driven system model, as well as their descriptions and values.

| Description | Parameter | Value | Unit |
|---|---|---|---|
| Moment of inertia | $J_m$ | 10.27 | kg·m$^2$ |
| Armature resistance | $R_a$ | 1165.2 | Ω |
| Torque coefficient | $K_t$ | $7.3892 \times 10^6$ | N·m/A |
| Viscous friction coefficient | $B_m$ | 6.474 | N·m·s/rad |
| Back electromotive force coefficient | $K_b$ | 0.0294 | V·s/rad |
| Total of worktable mass | $M_t$ | 7 | kg |
| Coefficient of the damping of the lead screw | $C_t$ | 10566 | N·s/m |
| Coefficient stiffness of the lead screw | $K_s$ | $5.18 \times 10^6$ | N/m |
| Coefficient of motor rotation converts to lead screw | R | 0.7958 | - |

## 2.2. Fault Tolerant Control by the Artificial Neural Network (ANN) based on the PI Servo System and Observer

The fault tolerant control presented in Figure 7 is combined with three portions of the system: the controller loop, fault detection and diagnostic scheme, and gain compensation module. The observer is used to estimate the state variable for use in feedback to the control loop instead of for measurement by the sensor. The observer error is used to enable feedback to the fault detection and diagnostic module for classification by ANN and to select the appropriate estimate gain ($K_f$) to compensate back to the controller to make the system run under the desired conditions. The design and development of the PI servo system [11] in this article are shown in Figure 7. To track and determine the transience of the response signal and support the fault tolerant control architecture, the stability criteria must be analyzed early in the process. Thus, the pole placement technique was used to assign the state feedback gain ($K$) and PI servo controller gain ($K_i$). The design structure requires state variable feedback using the gain $K$. In practice, the measurement of all state variables is difficult, and values are not discernible from each other. The observer design approach was used to estimate state variables under the measured output signals and measured inputs.

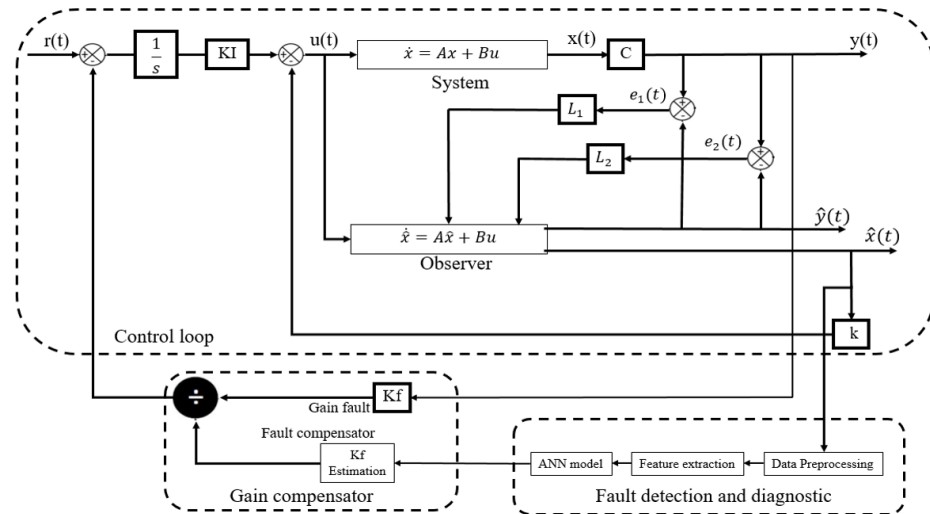

**Figure 7.** Fault tolerant control architecture.

When using control design methods, the first step must be to check the controllability and the observability [21] of the system. For the system to be completely controllable, the matrix shown below was utilized:

$$P_c = [\ \mathrm{B} \quad \mathrm{AB} \quad \mathrm{A}^2\mathrm{B} \quad \dots \quad \mathrm{A}^{n-1}\mathrm{B}\ ]$$

$$= \begin{bmatrix} 9.4823 \times 10^3 & -4.5907 \times 10^6 & 2.2149 \times 10^9 & -1.068 \times 10^{12} & 515.5849 \times 10^{12} \\ 0 & 0 & 28.473 \times 10^3 & -13.825 \times 10^6 & 6.670 \times 10^9 \\ 0 & 28.473 \times 10^3 & -13.825 \times 10^6 & 6.6704 \times 10^9 & -3.2183 \times 10^{12} \\ 0 & 0 & 0 & 0 & 582.089 \times 10^9 \\ 0 & 0 & 0 & 582.089 \times 10^9 & -3.498 \times 10^{15} \end{bmatrix}$$

This shows the determinant of the matrix; i.e., that the system is controllable. The observability can be examined from the matrix $P_0$, given by:

$$P_o = \begin{bmatrix} C \\ CA \\ \vdots \\ CA^{n-1} \end{bmatrix}$$

$$= \begin{bmatrix} 0 & 0 & 0 & 1 & 0 \\ 0 & 0 & 0 & 0 & 1 \\ 0 & 20.443 \times 10^6 & 0 & -25.558 \times 10^6 & -5.524 \times 10^3 \\ 0 & -112.930 \times 10^9 & 20.443 \times 10^6 & 141.189 \times 10^9 & 4.957 \times 10^6 \\ 61.387 \times 10^6 & 101.348 \times 10^{12} & -112.958 \times 10^9 & -126.708 \times 10^{12} & 113.802 \times 10^9 \end{bmatrix}$$

It was found that the determinant of the matrix was $P_0 \neq 0$, which is equal to the order of the system. This confirms the observability of the system. The next step to finding the state of feedback gain was to determine the location of the closed-loop pole in order to select the position of the poles with direct consistency with the eigenvalues of the system, controlling the property of the response of

the system. Ackermann's formula is a supplementary method written in Equations (6)–(8). From the Cayley–Hamilton theorem and the characteristics equation, we obtain the following:

$$\phi(A) = B(\alpha_2 K + \alpha_1 K\widetilde{A} + K\widetilde{A^2}) + AB(\alpha_1 K + K\widetilde{A}) + A^2 BK$$
$$= [B|AB|A^2 B]\begin{bmatrix} \alpha_2 K + \alpha_1 K\widetilde{A} + K\widetilde{A^2} \\ \alpha_1 K + K\widetilde{A} \\ K \end{bmatrix} \tag{6}$$

$$K = [\begin{matrix} 0 & 0 & \cdots & 0 & 1 \end{matrix}][B|AB|\cdots A^{n-1}B]^{-1}\phi(A) \tag{7}$$

where the desired characteristic polynomial of the closed-loop pole can be given by

$$s^n + \alpha_1 s^{n-1} + \alpha_2 s^{n-2} + \ldots\ldots + \alpha_{n-1}s + \alpha_0 = 0 \tag{8}$$

The design of the clamping unit position control provides an overdamped response. Since the augmented matrices had a form of $5 \times 5$, we placed the closed-loop poles at $-2$ and $-4$ and add the fourth pole, which has five times the value of the prominent pole in the system, which is $-10$. All status vectors are estimated using observers. From the system design, as shown as Figure 6, the state space can be defined by [21]:

$$x = Ax + Bu \tag{9}$$

$$y = Cx \tag{10}$$

The equation of the observer is roughly the same as that of the system, except for the addition of a term which includes the estimation error to compensate for inaccuracies in metrics A and B.Th. The estimation error or observer error is the difference between the measured output and the estimated output. Thus, we define the mathematical model of the observer to be:

$$\hat{x} = A\hat{x} + Bu + L_1 C[y - C\hat{x}] + L_2 C[y - C\hat{x}] \tag{11}$$

where $L_1 = L_2 = L$.

$$\hat{x} = A\hat{x} + Bu + 2LC[y - C\hat{x}] \tag{12}$$

$$\hat{x} = A\hat{x} + Bu + 2LC[x - \hat{x}] \tag{13}$$

where $\hat{x}$ is the estimated state, $L_1$ and $L_2$ are the observer gain for the linear encoder and rotary encoder, respectively, and $C\hat{x}$ is the estimated output. Consider the system by subtracting Equation (9) and Equation (13); thus, the error equation is presented as Equation (14):

$$x - \hat{x} = Ax - A\hat{x} - 2LC[x - \hat{x}]$$

$$e = Ax - A\hat{x} - 2LC[x - \hat{x}]$$

where

$$e = (x - \hat{x}), e = (x - \hat{x})$$

Thus:

$$e = [A - 2LC]e \tag{14}$$

Equation (14) demonstrates the dynamic behavior of the error between the observer and plant. The eigenvalue of a matrix $[A - 2LC]$ defined for the matrices of observer gain showed that the observer pole is 10 times faster than the closed-loop pole.

### 2.3. Data Manipulation

Data manipulation is used to process the raw data from the observer error signal, and then to extract the mean value by the feature extraction process; these experiments were investigated for 11 linear fault conditions, referring to the experiment setup flow. The mean value of each observer error signal is given as the input for ANN, both in terms of pattern recognition and model fitting [14]. The input data were divided into training (70%), validation (15%), and testing (15%), respectively, as shown in Figure 8.

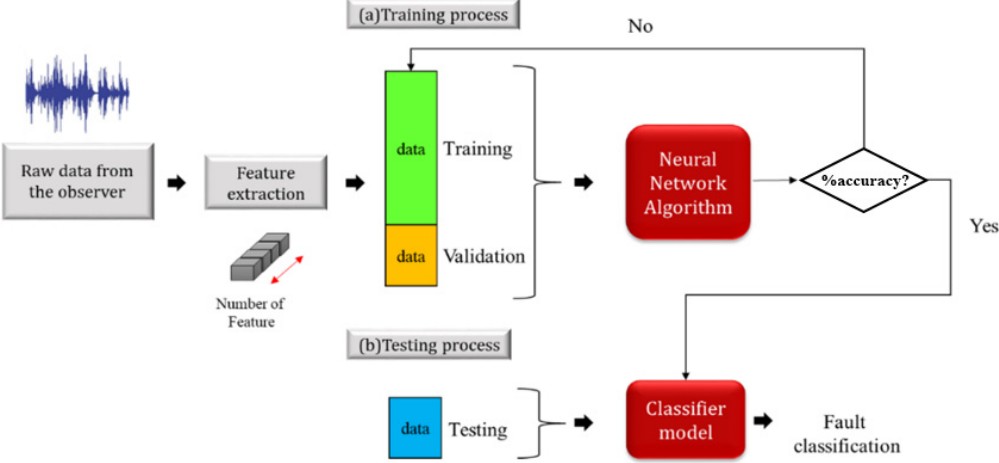

**Figure 8.** The procedure flow of the classification model.

### 2.4. Artificial Neural Network (ANN)

The artificial neural network (ANN) is a common machine-learning tool which is widely used in data analysis. This work used two types of ANN: model fitting and pattern recognition. Both methods use a set up of three layers—the input layer, hidden layer and output layer—as shown in Figure 9a,b. The input layer contains the observer error signal with 11 conditions; for each condition, 50 data sets are collected, making the total of data sets for input equal to 550. The data are divided into 75% for training and 15% for validation and testing, respectively. The hidden layer was set up with 50 layers to minimize the training time. The last layer is the output layer, which consists of 11 target outputs. Both ANN analysis techniques were processed by the MATLAB application. In this work, the ANN pattern recognition, as shown in Figure 9a, utilized a scaled conjugated gradient (*trainscg* in MATLAB®) for the learning algorithm, as this learning algorithm requires less memory. The ANN model fitting is shown in Figure 9b and used the Levenberg–Marquardt algorithm (*trainlm* in MATLAB®). This algorithm requires more memory compared to a scaled conjugated gradient but less time. The ANN pattern recognition classifies the sensor fault and integrates it with the gain compensation by discrete gain scheduling, while the model fitting estimates the gain value for compensation by continuous gain scheduling.

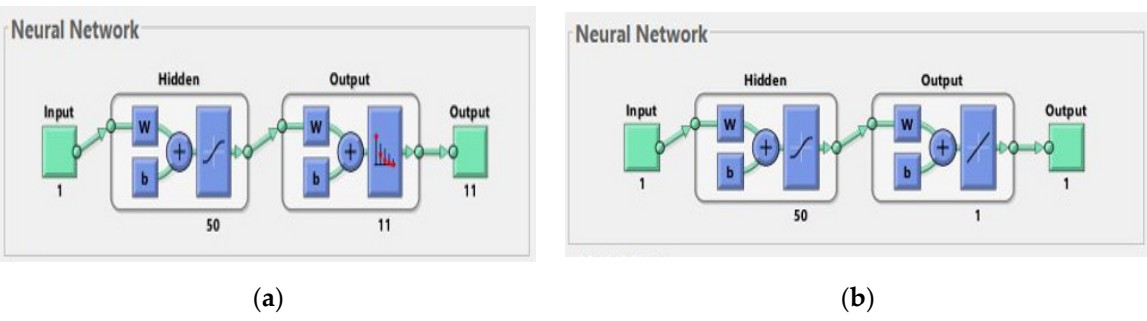

(**a**)        (**b**)

**Figure 9.** (**a**) Pattern recognition neural network training; (**b**) model fitting neural network training.

### 2.4.1. ANN Pattern Recognition

ANN pattern recognition is the process of classifying data into a group format according to the characteristics of those data. To illustrate the differences between groups of data, we classify data groups by using statistical tools to aid in feature extraction and use training data to train the system to determine which data belong to the same data group. There are several models for classifying data, such as decision tree, mathematical formulae, classification (If–Then) rules or artificial neural network pattern recognition. The raw data are utilized to train the model, and the remaining data are used to validate and test the model, as shown in Figure 10. The classification model is a supervised model, which requires a target or variable to measure the prediction of the model. The model learns by the target of the classification; therefore, the classification model can measure the accuracy of the prediction (accuracy) by using the confusion matrix, as shown in Figure 11.

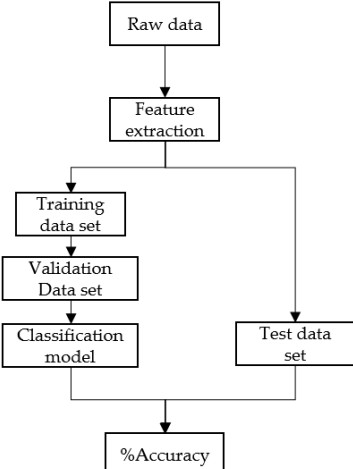

**Figure 10.** Artificial neural network (ANN) pattern recognition flow.

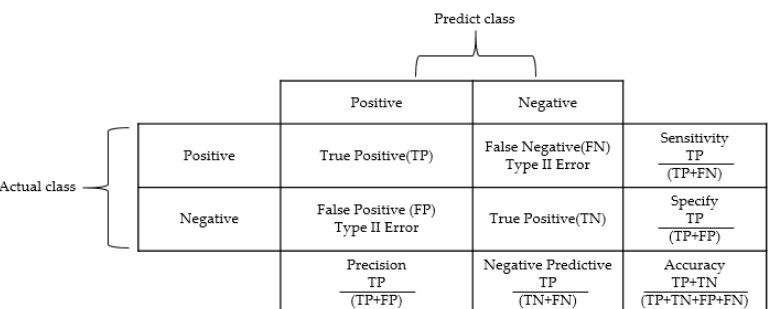

**Figure 11.** Confusion matrix.

The confusion matrix [22] is used to describe the accuracy of the model classification; the rows correspond to the predicted class (output class), and the columns correspond to the true class (target class). The diagonal cells demonstrate that the observations are correctly classified. The accuracy was calculated using Equation (15):

$$Accuracy = \frac{TP + TN}{TP + TN + FP + FN} \tag{15}$$

where *TP* shows that the observation is positive and predicted to be positive; *TN* shows that the observation is negative and predicted to be negative; *FP* shows that the observation is negative but predicted as positive; and *FN* shows that the observation is positive but predicted as negative [23].

### 2.4.2. ANN Model Fitting

Model fitting is the analysis of the relationships between the dependent (target) and independent variable (predictor) depicted by Figure 12 by using the best fit straight line. It is presented by Equation (16):

$$y = a + bx + e \tag{16}$$

where $b$ is the slope of the line and $e$ is an error term. This equation is used to predict the value of the target variable by using the predictor variable. The regression R squares measure the correlation between outputs and targets. An R square value close to 1 shows a good relationship.

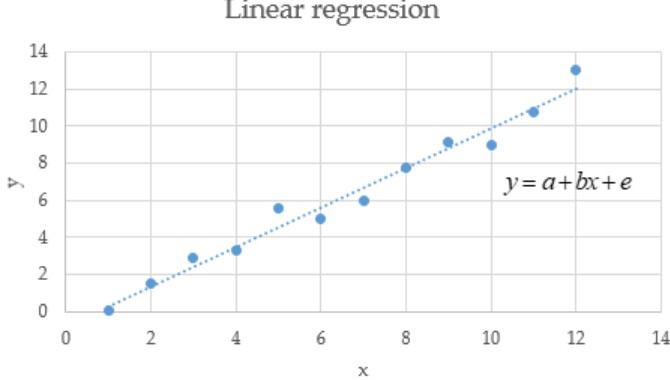

**Figure 12.** Linear regression.

The process of the model fitting is shown in Figure 13; for feeding the set of data for training and then selecting the learning algorithm, this study selected the regression algorithm. The system used Equation (16) to fit the model.

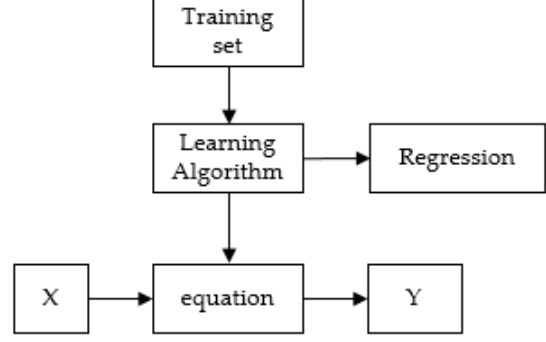

**Figure 13.** Process flow of model fitting.

### 2.5. Gain Scheduling

The gain-scheduling control method is the design of the controller and is achieved by examining the operating conditions of the process and then determining the appropriate gain setting and adjusting the controller parameters to maintain the system operation with the desired conditions, as shown in Figure 14. The control system is adjusted using forwarding compensation. The traditional concept of gain-scheduling is the application of flight control systems by measuring the Mach number and the dynamic pressure by sensors, which are used as scheduling variables. With timing variables, the controller parameters are calculated according to the number of operating conditions using the appropriate design method. System stability and performance are generally evaluated by simulation. The drawback of scheduling is that it is an open-loop compensation; there is no feedback to compensate for incorrect scheduling, and the design may take a long time. The advantage of scheduling in advance is that controller parameters can change quickly in response to process changes.

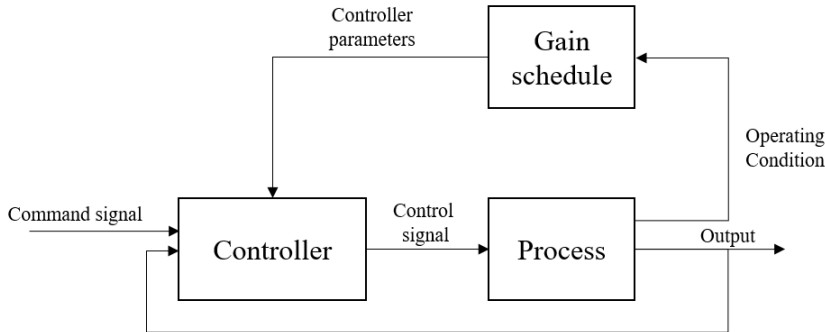

**Figure 14.** Gain scheduling controller design.

This work is designed with two types of gain compensation: discrete gain scheduling and continuous gain scheduling.

### 2.5.1. Discrete Gain Scheduling

The design of this discrete schedule is done using an experiment to collect the data of the observer error range. By varying the gain error by scaling the gain ($K_f$) to the system from −1.0% to 1.0 % with 0.2% increments and creating the matrix as shown in the table, the gain compensator ($K_f$) is also created, as shown in Table 2, to compensate for the gain error.

**Table 2.** Gain scheduling table.

| Gain Error (*Kf*) | Observer Error Range (mm.) | | Gain Compensation (*Kf Estimate*) |
| --- | --- | --- | --- |
| | Min | Max | |
| −1.00% | −0.0437 | −0.0381 | 0.990 |
| −0.80% | −0.0358 | −0.0232 | 0.992 |
| −0.80% | −0.0358 | −0.0232 | 0.992 |
| −0.60% | −0.0171 | −0.0128 | 0.994 |
| −0.40% | −0.006 | −0.002 | 0.996 |
| −0.20% | 0.0047 | 0.0098 | 0.998 |
| 0.00% | 0.0167 | 0.0205 | 1.000 |
| 0.20% | 0.0276 | 0.032 | 1.002 |
| 0.40% | 0.0396 | 0.0438 | 1.004 |
| 0.60% | 0.051 | 0.0556 | 1.006 |
| 0.80% | 0.0599 | 0.0656 | 1.008 |
| 1.00% | 0.0733 | 0.0779 | 1.010 |

### 2.5.2. Continuous Gain Scheduling

Continuous gain scheduling uses the information from the ANN model fitting to estimate the gain (*Kf*) based on the linear Equation (16) to compensate for the appropriate gain value to the system.

## 3. Experimental Setup

The experiment setup used linear state actuators of the ACAM machine, as shown in Figure 15, which was designed and fabricated as a simulation module to demonstrate and simulate control systems with a fault tolerant control scheme. The main components included a THK lead screw with a 5-mm pitch diameter, which was driven by a 200 watt Mitsubishi DC servo motor with a 2500 PPR incremental encoder and a 24-volt power supply. The operating speed of the motor was from 1200 to 1600 RPM for system identification. The Renishaw linear encoder model RGH22Z30D00 was used for feedback on the position of the worktable. An Omron E6B2-CWZ1X rotary encoder was used to double-check the position of the clamping unit. The motor driver amplifier received the control signal from the control unit and then fed this to the controller of the DC servomotor. The RAPCON platform controller interface was implemented together with the MATLAB/Simulink package to design

the proposed method and use it for data collection. This work used dual encoders to evaluate the observer's estimation of error results.

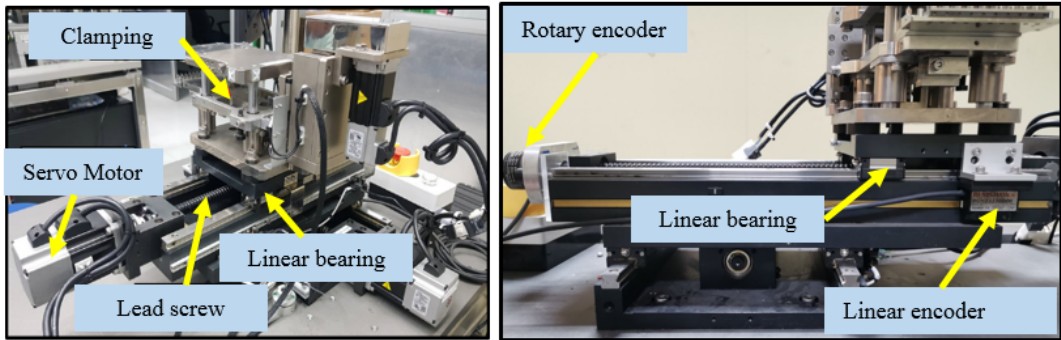

**Figure 15.** XY stage actuator.

The experiment was conducted as shown in Figure 16, with four stages: the first stage was to set up the work table and perform the simulation by scaling the gain by −1% to 1% with an increase of 0.2% for each condition, with the total fault condition equal to 10 and 1 representing the healthy condition. In the second stage, 50 data sets per condition were collected; the data were pre-processed and fed into the ANN process in stage three for modeling and the validation of the model. The final stage was the evaluation of the performance of the fault tolerant control of both gain compensation methods.

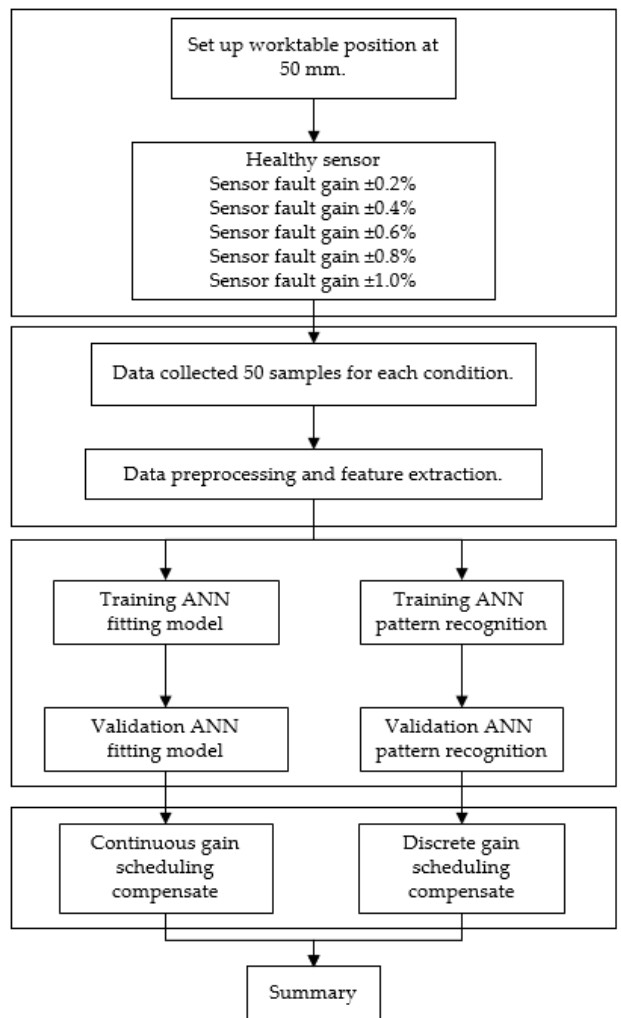

**Figure 16.** Process flow of the experiments.

## 4. Results and Discussion

### 4.1. Response Tracking and Observer Performance

The PI servo system with state estimation was used to design the controller, the observer was validated by response tracking, as shown in Figure 17, and the estimated accuracy of the output was compared to the actual output. The control position of the work table was tested for the input design at 8 mm. The pitch distance refers to the actual sequence of the ACAM machine. It was found that the method aiming to control the position could track the reference inputs, which could be compensated and could minimize errors for the desired step response. Then, the work table was set up for the experiment by moving it 50 mm, as shown in Figure 18.

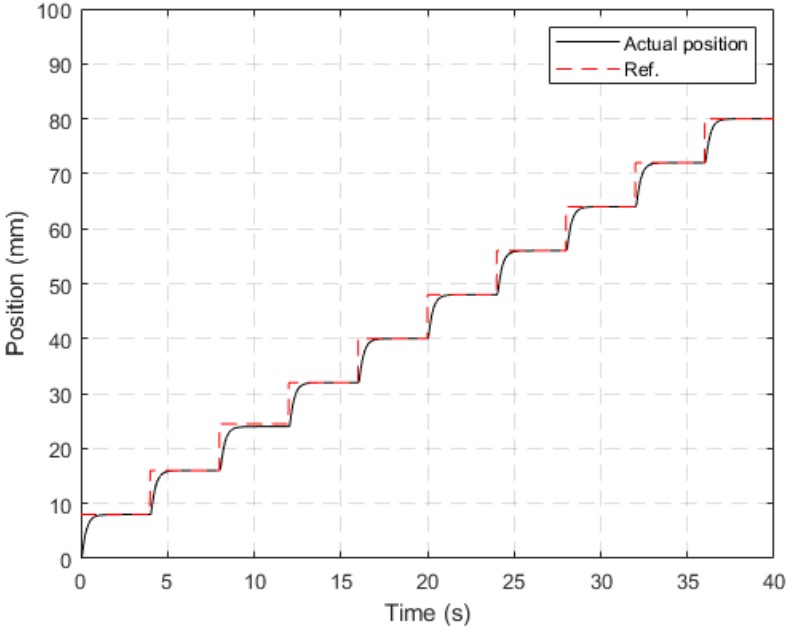

**Figure 17.** Tracking response of the feed-driven system.

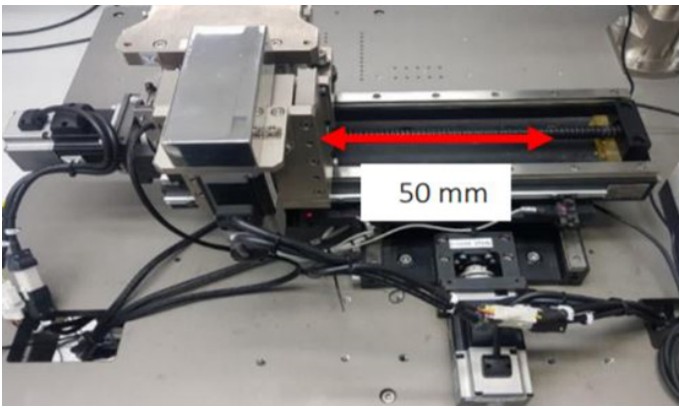

**Figure 18.** Worktable direction and position movement.

### 4.2. Data Collection and Preprocessing

The observer error signal information from the observer was collected by considering the transient response, as shown in Figure 19. Fifty data sets were collected for the 11 sensor fault conditions, making a total of 550 data sets. The mean value of the observer error was processed and fed-in to the ANN process.

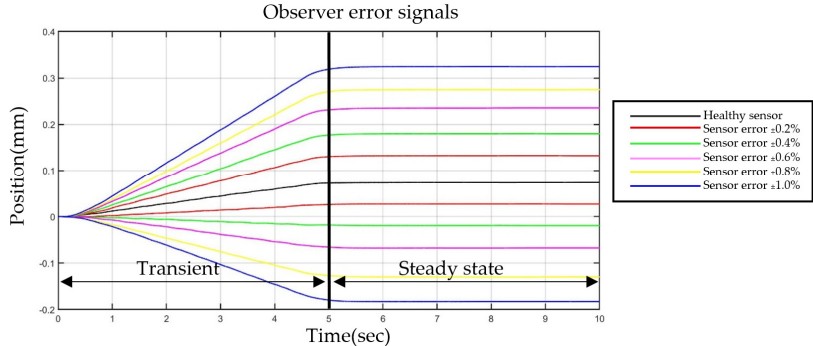

**Figure 19.** The observer error response.

### 4.3. Fault Detection and Diagnostic Experiment Result

### 4.3.1. Fault Detection by ANN Pattern Recognition

Fault detection was performed by the ANN pattern recognition process by dividing the observer error signal data into 70% (384 samples) for the training model, and 15% (83 samples) for validation and testing, respectively. The results were represented by the confusion matrix shown in Figure 20. All four confusion matrixes showed an overall accuracy equal to 100%. The results showed that the row confusion matrix was equal to 100% for all classes, and the column corresponds to the true class (target class), also showing 100% for all conditions. The result of the pattern recognition classification showed its effectiveness in classifying a linear sensor fault for all conditions.

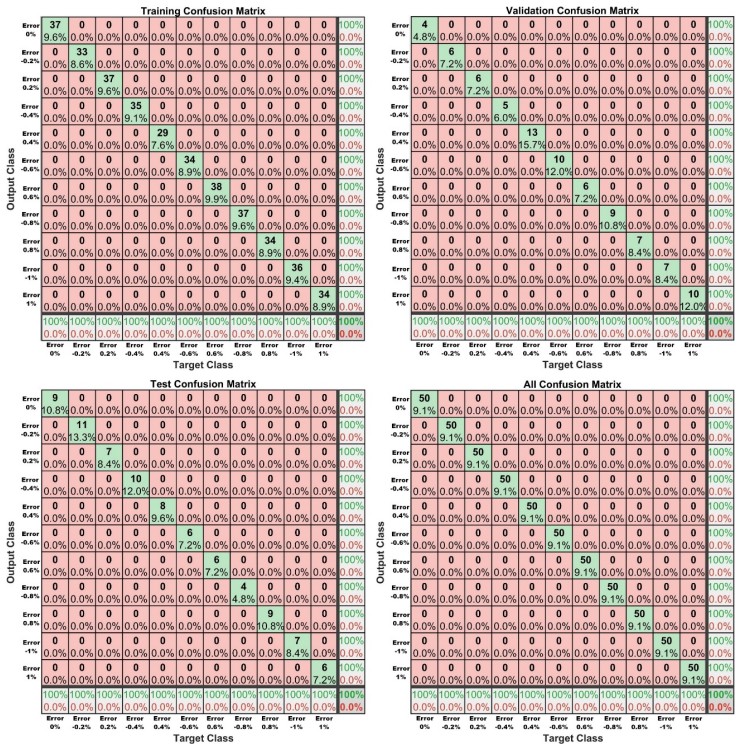

**Figure 20.** Neural network confusion matrix.

### 4.3.2. Fault Detection by ANN Model Fitting

The ANN model fitting displays the outputs concerning the targets for training, validation, and test sets. The R-square value was 0.9999 for all training, validation, testing and all sets, as shown in Figure 21, which depicts a reasonably good relationship between the output and target sets.

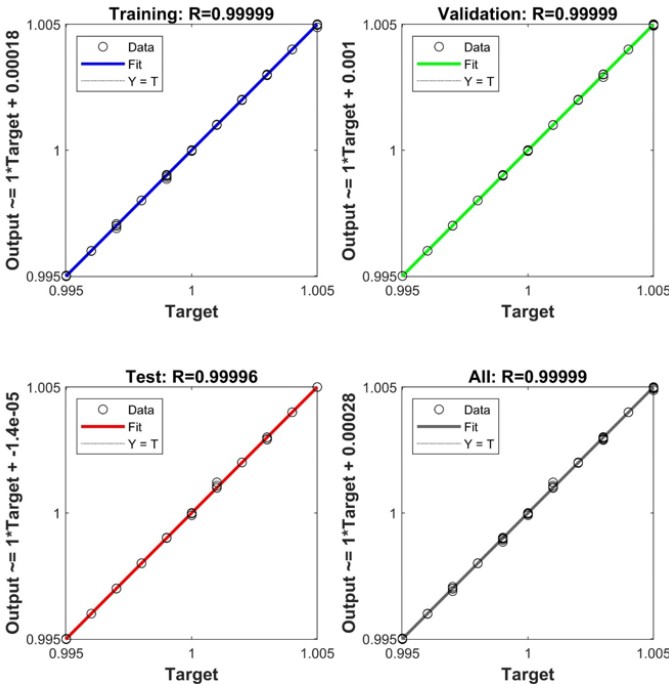

**Figure 21.** Neural network regression and function fitting of the ANN model.

In summary, the experimental results of the ANN for both pattern recognition and model fitting approaches are shown in Table 3. The accuracy of pattern recognition was 100%, and the R square value revealed a satisfying value of 99.99% for model fitting. The result shows that the ANN technique is effective in classifying the sensor fault condition. The classification result can be utilized as an input to improve the compensation process for the next experiment.

**Table 3.** Summary of fault detection and diagnostic by ANN.

| ANN Method | Accuracy/R-Squared |
|---|---|
| Pattern recognition | 100% |
| Model fitting | 99.99% |

### 4.4. Gain Compensation Experiment Result

The gain compensator is used to compensate for the appropriate gain value after detecting and classifying the sensor fault conditions by ANN from the previous experiment. The controller ensures that the system continues operation with satisfactory performance. The experiment used two groups: continuous gain scheduling and discrete gain scheduling. To validate the performance of gain compensation, the scaling gain errors were set up as follows: a healthy condition and 10 sensor fault conditions, followed by 0%, −0.2%, −0.3%, −0.5%, −0.6%, −0.9%,0.2%, 0.3%, 0.5%, 0.6%, and 0.9%. The results are described below.

#### 4.4.1. Result for Gain Compensation by Discrete Gain Scheduling

The result compares the setting position (50 mm) with the actual position between the systems with and without gain compensation. The gain estimation ($K_f$) from Table 2 is used for compensation after detection and diagnostics by the ANN pattern recognition. The result compares the three groups of data shown in Figure 22: the blue line is the setting position with 50 mm, the orange line is the response of the work table position for each sensor fault condition, and the grey line is the response after compensation by discrete gain scheduling. Table 4 is the summary result shows that the average of the position error was reduced from 0.214 mm to 0.031 mm. The result of the gain compensation

by discrete gain scheduling was to reduce the position error by 86% compared to the system without gain compensation.

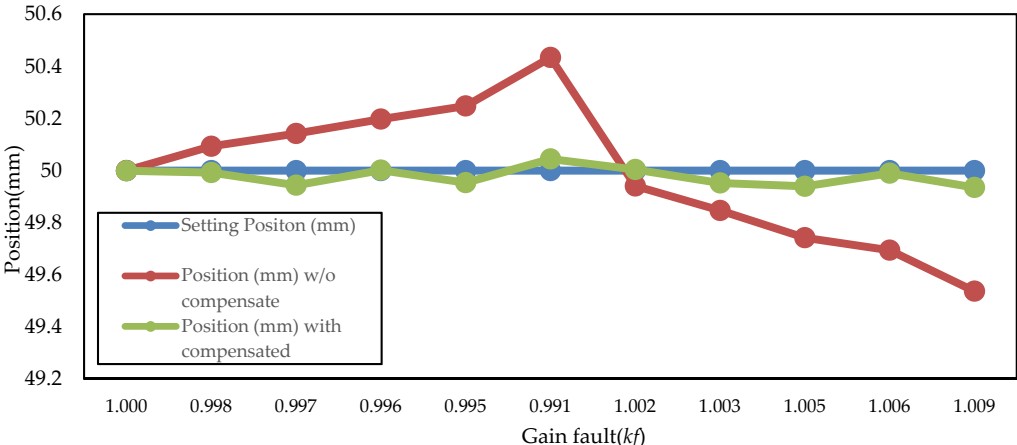

**Figure 22.** Gain compensation by discrete gain scheduling.

**Table 4.** Gain compensation result for discrete gain scheduling.

| Gain Fault *(Kf)* | Position (mm) without Compensation | Gain Estimate *(Kf)* | Position (mm) with Compensation | Position Error (mm) | |
|---|---|---|---|---|---|
| | | | | without Compensation | with Compensation |
| 1.000 | 50.000 | 1.000 | 49.999 | 0.000 | 0.001 |
| 0.998 | 50.093 | 0.998 | 49.992 | 0.093 | 0.008 |
| 0.997 | 50.142 | 0.996 | 49.944 | 0.142 | 0.056 |
| 0.996 | 50.198 | 0.996 | 50.002 | 0.198 | 0.002 |
| 0.995 | 50.248 | 0.994 | 49.953 | 0.248 | 0.047 |
| 0.991 | 50.434 | 0.992 | 50.044 | 0.434 | 0.044 |
| 1.002 | 49.941 | 1.002 | 50.004 | 0.059 | 0.004 |
| 1.003 | 49.847 | 1.002 | 49.952 | 0.153 | 0.048 |
| 1.005 | 49.743 | 1.004 | 49.940 | 0.257 | 0.060 |
| 1.006 | 49.694 | 1.006 | 49.990 | 0.306 | 0.010 |
| 1.009 | 49.537 | 1.008 | 49.935 | 0.463 | 0.065 |

### 4.4.2. Result for Gain Compensation by Continuous Gain Scheduling

The continuous gain compensator uses the linear regression Equation (17) to predict the gain value to compensate for the work table position and to sustain the system. Figure 23 compares the work table position of all three groups: the blue line is the setting position with a response of 50 mm, and the other two lines show the response of the work table without gain compensation and with gain compensation as orange and grey lines, respectively. The summary of the gain compensation result was shown in Table 5. The average position error of the worktable of the system without gain compensation was 0.228 mm, while the average position error of the system with gain compensation was shown to be 0.017 mm. The result was that the system with gain compensation was shown to be effective, reducing the position error by 93%.

$$Output \sim= 1xTarget + 0.00028 \tag{17}$$

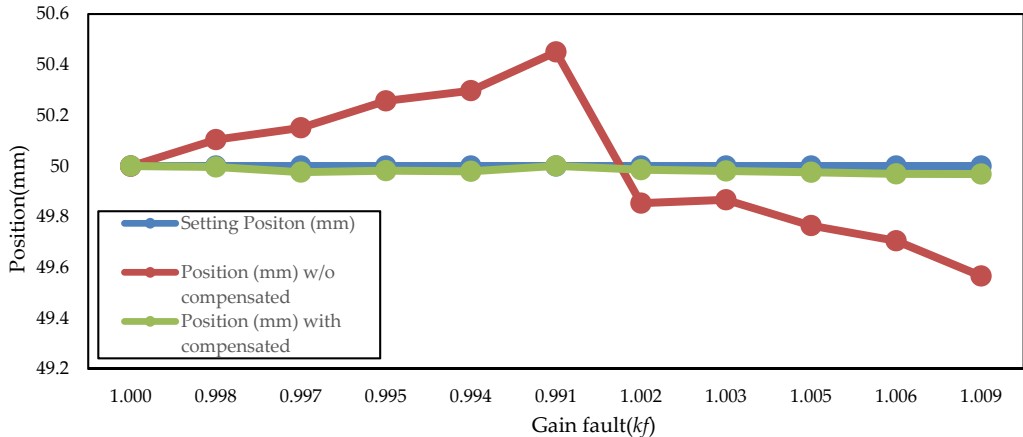

**Figure 23.** Gain compensation by continuous gain scheduling.

**Table 5.** Gain compensation result for continuous gain scheduling.

| Gain Fault *(Kf)* | Position (mm) without Compensation | Gain Estimate *(Kf)* | Position (mm) with Compensation | Position Error (mm) | |
| --- | --- | --- | --- | --- | --- |
| | | | | without Compensation | with Compensation |
| 1.000 | 50.000 | 1.000 | 50.000 | 0.000 | 0.000 |
| 0.998 | 50.104 | 0.998 | 49.996 | 0.104 | 0.004 |
| 0.997 | 50.151 | 0.997 | 49.976 | 0.151 | 0.024 |
| 0.996 | 50.257 | 0.994 | 49.982 | 0.257 | 0.018 |
| 0.995 | 50.298 | 0.994 | 49.979 | 0.298 | 0.021 |
| 0.991 | 50.450 | 0.991 | 50.001 | 0.450 | 0.001 |
| 1.002 | 49.854 | 1.001 | 49.986 | 0.146 | 0.014 |
| 1.003 | 49.867 | 1.002 | 49.981 | 0.133 | 0.019 |
| 1.005 | 49.765 | 1.004 | 49.976 | 0.235 | 0.024 |
| 1.006 | 49.706 | 1.005 | 49.970 | 0.294 | 0.030 |
| 1.009 | 49.567 | 1.008 | 49.969 | 0.433 | 0.031 |

The summary shown in Figure 24 and Table 6 shows that both gain compensation methods successfully reduced the position error by 86% and 93% for discrete and continuous gain scheduling, respectively.

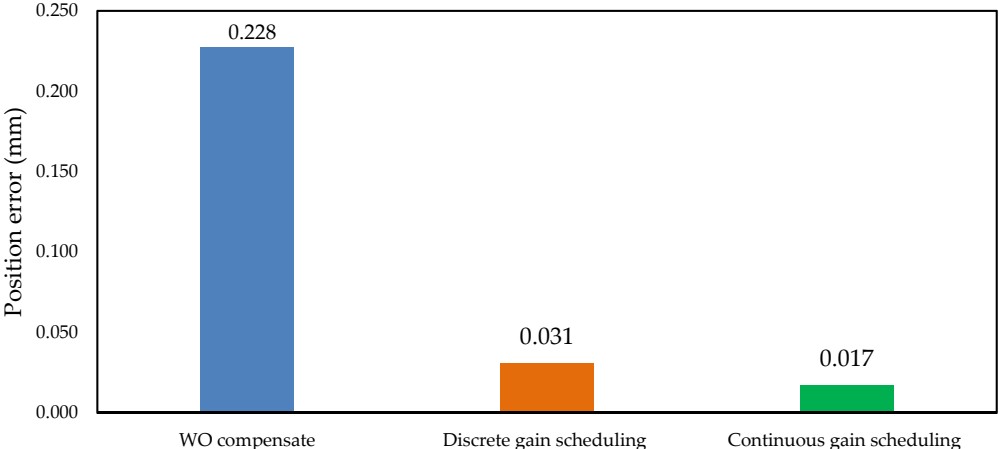

**Figure 24.** Comparison of systems with and without gain compensation.

**Table 6.** Summary of the gain compensation by varying the linear gain.

| Condition | Position Error (mm) | % Position Error Reduction |
|---|---|---|
| Without compensate | 0.228 | - |
| Discrete gain scheduling | 0.031 | 86% |
| Continuous gain scheduling | 0.017 | 93% |

## 5. Conclusions

This work presented the fault tolerant control of a high-speed auto core adhesion mounting machine based on the PI servo design of the observer. The approach combined fault detection and diagnostics by ANN with the design of a modern control system and a gain compensation technique. The experiment included three investigations: the first was tracking the performance of the PI servo controller with state variable estimation; the second was the creation of a fault detection and diagnostic model, simulated with 11 fault conditions of the linear encoder, and the state variable with observer error was extracted to the mean statistical features for the training data set; and the final step was gain compensation by continuous and discrete compensation methods. The following conclusions can be drawn from our work:

- The tracking response of the controller by the PI servo system with state estimation based on an observer was found to provide effective enhancement in position control and was able to track reference inputs, which compensated and significantly reduced errors, leading to the desired step response.
- For the fault detection and diagnostics of linear encoder faults by the ANN pattern recognition and model fitting, by using the observer error signal from the observer, the approaches successfully classified the sensor fault condition with an accuracy of 100% for the pattern recognition method and an R-square of 99.99% for the model fitting technique.
- Both gain compensation techniques—continuous gain scheduling and discrete gain scheduling—were shown to successfully compensate the gain value to maintain the position error of the worktable, moving it back to the desired position, as shown as Table 6. With discrete gain scheduling, position error was reduced from 0.228 mm to 0.031 mm (86% reduction), while the continuous gain scheduling reduced the error from 0.228 mm to 0.017 mm (93% reduction) compared with the system without gain compensation.
- Fault tolerant control based on PI servo design with an observer by using the ANN and gain compensation technique exceeded the process requirements in controlling the position of the worktable, maintaining the suspension reference hole position within the FOV for slider attachment and the adhesive dispensing process.

**Author Contributions:** Conceptualization, P.C. and J.S.; methodology and software, P.C. and T.W.; validation, P.C. and J.S.; formal analysis, P.C.; investigation, J.S; writing—original draft preparation, P.C.; writing—review and editing, J.S. and S.T. All authors have read and agreed to the published version of the manuscript.

**Funding:** This research was funded by the Research and Researchers for Industry (RRI) program under the Thailand Research Fund (TRF). It offers a scholarship for Ph.D. Students, Suranaree University of Technology (SUT) agency through grant number PHD60I0043.

**Acknowledgments:** The authors are indebted to Suranaree University of Technology (SUT) and Western Digital (Thailand) Company Limited for their generosity and valuable comments.

**Conflicts of Interest:** The authors declare no conflict of interest.

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
