# Peer review of "Fault Tolerant Control Based on an Observer on PI Servo Design for a High-Speed Automation Machine"

_machines, doi:10.3390/machines8020022_

Round 1

Reviewer 1 Report

  1. English revision: “15 fault”, “35 linear bearing”, “38 requirement”, “an 63”, “65 accuracy”, “88 diagnostic”, etc.;
  2. The bibliography is limited (12 references);
  3. Citation is missing from important parts of this paper, e.g., many equations, ANN, Conjugated Gradient Algorithm, Levenberg-Marquardt, confusion matrix, etc.;
  4. Citations from benchmark publications in the specialized literature are missing;
  5. I suggest replacing “Artificial Neuron Network” with “Artificial Neural Network”;
  6. Define acronyms only once in the article, just the first time they are indicated;
  7. Line 145: This matrix has strange characters, and also in equation (10);
  8. Figure 7: Replace “% accuracy” with “% accuracy ?”;
  9. There is a need to present the details of the neural network: structure, parameters, training technique, etc .;
  10. The presentation of Figure 8 is unnecessary because it does not offer any important information;
  11. Replace “sec” by “s” (SI – Système International d'unités);
  12. The data referring to the experiment, mainly in relation to ANN, must be made available in order to reproduce the results if the reader wishes;
  13. I think that this article can be perfectly reduced, mainly in relation to the presentation of the experiment;
  14. I suggest to the Authors to highlight (objectively) the innovation of this proposal in relation to literature.

Author Response

Dear Reviewer I

Thank you very much for your comments and suggestions. I has revised and rewrite per your suggestion detail of each topic please find in as attached file, and I have set the Track change for you can check all the item that I have revised.

Best regards

Srisertpol J. 

Reviewer 2 Report

The authors claimed to present a new technique for fault tolerance control. Although the proposed ideas are interesting, with an apparent contribution of the study, there are many issues that negatively impact the paper and that should be considered in detail.

The main ideas in the paper unfortunately are not very well presented. Sentences are not objective or correctly linked, lacking in organization and clarity. Paragraphs are too long and issues, as incorrect use of grammar and typos, are very common in the text. English errors are common for non-native speakers, but in this case they make it difficult to follow the stream of thoughts presented by the authors, with a significative impact on the overall quality of the paper. 

There are several other problems regarding the form of the paper. All figures have inadequate resolution. In some cases, it even affects the understanding of such figure (the confusion matrices in Fig. 16, for instance, are very hard to read). Additionally, there are non-vectorized graphs (e.g. Fig. 18) and print-screen images (e.g. Fig. 17), which are not suitable for publication.

The presence of unknown characters in some equations (e.g. Eq. 10) compromises the quality of the paper as it hinders the understanding of the methodology followed by the authors.

Among other problems regarding content, there are some inconsistences in the methodology (amplified by the uncoherent text and equation problems) and a shallow bibliographical research (some important papers in the area are missing).

These aforementioned problems, at my point of view, greatly weakens the manuscript proposed by the authors. Therefore, I suggest that the authors rewrite (almost entirely) the manuscript.

Bearing these facts in mind and considering the quality of the Machines journal, I am convinced that the present paper is not acceptable for publication as it is.

Author Response

Dear Reviewer II
Thank you very much for your comments and suggestions. I has revised and rewrite per your suggestion detail of each topic please find in as attached file, and I have set the Track change for you can check all the item that I have revised.
Best regards
Srisertpol J.

Reviewer 3 Report

A new method  for fault tolerance control base in which An Artificial Neural Network is used  to validate and compare the performance of fault detection is proposed.

The authors should highlight in the introductory section how the proposed technique aims to improve the performance of the FTC and which performance improves.

The PC matrix shown at row 145 is not clearly visible, illegible symbols appear inside the matrix.

Illegible symbols are also present in the equation at row 170.

The flowchart  of the classification model in Fig. 7 should be discussed in more detail.

Rather than presenting the simple percetron model, authors should motivate the choice of their neural model. It is not necessary to describe generically an ANN, well known in the literature, but it is fundamental to explain which type of ANN is adopted, how many hidden layers it needs, how many nodes are needed for the hidden layers depending on the input nodes, what is the complexity computational of the network in relation to the performance level to be achieved.

Author Response

Dear Reviewer III

Thank you very much for your comments and suggestions. I has revised and rewrite per your suggestion detail of each topic please find in as attached file, and I have set the Track change for you can check all the item that I have revised.

Best regards

Srisertpol J.

Round 2

Reviewer 2 Report

The answers and modifications provided by the authors after revision have improved significantly the paper.
In my opinion, it is now suitable for publication.

Author Response

Dear Reviewer 

Thank you very much for all of you review and comments. I has revised and undergone English language editing by MDPI, detail of each action please find in as below, and I have set the Track change for you can check all the item that I have revised

Best regards

Srisertpol J.

Reviewer 3 Report

The author has taken all my suggestions into consideration, greatly improving the quality of this paper. I considered this paper to be published in its current form.

Author Response

(The authors gave the same response as above.)
